# Generation of High-Resolution Gridded Runoff Product for the Republic of Korea Sub-Basins from Seasonal Merging of Global Reanalysis Datasets

Woo-Yeon Sunwoo [1], Hoang Hai Nguyen [2] and Kyung-Soo Jun [1,*]

1   Water Resources Engineering Lab, Department of Water Resources, Graduate School of Water Resources, Sungkyunkwan University, Suwon 440-746, Republic of Korea; swwy@skku.edu
2   Sejong Rain Co., Ltd., In-House Venture of K-Water, 99 Daehak-ro, Yuseong-gu, Daejeon 34134, Republic of Korea; hai.nh1812@gmail.com
*   Correspondence: ksjun@skku.edu; Tel.:+82-312907642

**Abstract:** Gridded runoff product at the sub-basin level is pivotal for effective hydrologic modeling and applications. Although reanalyses can overcome the lack of traditional stream gauge networks to provide reliable geospatial runoff data, the inherent uncertainties associated with single products are still a problem. This study aims to improve the single products' limitations over the heterogeneous Republic of Korea region by merging three common global reanalysis datasets to generate a high-quality and long-term gridded runoff product at a high resolution. The merging method relies on triple collocation (TC) analysis, which requires no reference runoff dataset, with a modification that was applied separately to wet and dry seasons (seasonal merging). A comparison between the merged runoff and its parent products at 0.10° grid, on a daily basis, and using the entire 10-year period (2011–2020) against an independent ground-based sub-basin runoff product generally indicated a superior performance of the merged product even at the national scale of Republic of Korea. Moreover, a slight improvement obtained with the seasonal merging compared to the traditional all-time merging highlighted the potential of this modification to address several drawbacks in the TC assumption, especially the non-stationary runoff pattern caused by seasonal rainfall effects in the Republic of Korea. Despite the need for further improvement such as bias correction, the results of this study encourage making a reliable benchmark runoff product at a regional scale, which is beneficial for flood/drought monitoring and artificial intelligence-based hydrologic model training.

**Keywords:** runoff; merging; triple collocation; seasonal; reanalysis; sub-basin

## 1. Introduction

Total runoff, which encompasses both surface runoff and baseflow, represents overall water flowing over land surfaces and through subsurface pathways before entering rivers or water bodies via discharge. Since runoff plays a vital role in the water cycle and climate sectors, accurately capturing its spatiotemporal dynamics at relevant scales can enhance hydrologic modeling [1] and applications such as water resources and agriculture management [2], weather and climate forecasting [3], flood and drought monitoring [4], or dams and hydropower plants operation [5]. Moreover, alongside the growth of big data and artificial intelligence when data-driven machine learning/deep learning models show great potential to replace traditional process-driven hydrologic models [6], long-term and high-quality geospatial runoff estimates can serve as a reliable benchmark for training artificial intelligence-based hydrologic models over large areas.

Although the most reliable streamflow or river discharge data are accessible from a wide range of available stream gauge networks located near river outlets, their low density and lack of in situ runoff networks over land can make the generation of large-scale gridded

runoff product directly from those point-scale in-situ observations become more challenging [7–9]. In addition, high spatial heterogeneity caused by combined effects of key factors including topography, land use and land cover, soil and vegetation, hydrometeorological variables, or anthropogenic activities also introduces large uncertainties in the interpolated geospatial runoff products from point measurements [9–14].

Reanalyses offer an alternative way to overcome the aforementioned limitations of ground-based networks for generating reliable long-term continuous gridded runoff data by integrating multiple data sources (in situ and satellite observations and model forecasts) via their data assimilation systems. Various global, gridded, and continuous runoff products ranging from surface flow to baseflow for runoff types, from hourly to monthly for temporal resolution, and from medium scale to coarse scale for spatial resolution, can be provided by common reanalyses. Among them, the typical products are the European Centre for Medium-Range Weather Forecasts (ECMWF) ReAnalysis v5 (ERA5) [15] and its land version (ERA5-Land) [16]; or the Global Land Data Assimilation System (GLDAS) [17] and the Modern-Era Retrospective analysis for Research and Applications (MERRA) [18] from the National Aeronautics and Space Administration (NASA). However, the major limitation of a single reanalysis product lies in the inherent uncertainties associated with its different forcing datasets and construction methodologies used, suggesting the need for merging multiple reanalysis outputs to overcome the single product's problem. In the context of reanalysis-based runoff products, even though the most superior performance related to the global synthesized runoff product was reported in the literature [19], observed streamflow data were often required as the reference for characterizing error information on each single model's output, which can be then used to compute the corresponding weights when merging those multiple sources.

Many efforts have been made to develop merging techniques with less dependence on a true reference product. Among them, triple collocation (TC) analysis is placed as one of the top priority methods, which allows the error characterization and combination of multiple standalone and collocated datasets without any requirement of truth reference datasets [20,21]. This technique has been widely applied for global geospatial data in the hydrology sector, mainly focusing on precipitation [22–24], soil moisture [25–27], and evapotranspiration [28–30], but it has hardly ever been employed in the field of runoff. Another problem is that the TC merging method was preferred at the global scale rather than at the sub-basin level, while the runoff can strongly vary even at a small scale across adjacent sub-basins due to the high spatial heterogeneity of the landscape. In the context of the Republic of Korea where diverse vegetation conditions, complex terrains, and strong seasonal rainfall differences between rainy (wet) and non-rainy (dry) seasons under the summer monsoon's (Changma) effects [31] may lead to high uncertainties in runoff behavior, focusing on improving seasonal runoff estimates at the sub-basin level with more detailed spatial information derived from global products is necessary [32–36].

Overall, this study aims to develop a high-quality and long-term continuous gridded total runoff product at the high resolution generated from merging several single widely used reanalyses over the Republic of Korean sub-basins, towards supporting future studies as a reliable benchmark product for flood and drought monitoring, hydrologic simulation evaluation, data-driven rainfall-runoff model development, or climate change adaptation in this region. In particular, three commonly used reanalyses including the ERA5-Land (ERA5L), GLDAS, and MERRA were considered, with a targeted spatial resolution of 0.1° grid (~10 km, provided by the highest spatial resolution product, the ERA5L) and a targeted temporal resolution of daily data. The TC method was employed as the major merging method to generate the synthesized runoff product from three selected reanalyses' outputs. Nevertheless, a slight advance of this study compared to previous studies of TC merging is that the technique was applied separately for wet and dry seasons, expecting to enhance the seasonal runoff estimates. We compare the performances when applying seasonal and all-time merging to investigate whether seasonal TC merging is suitable for improving runoff estimates at the sub-basin level over the Republic of Korea region.

## 2. Study Area, Datasets, and Methods

### 2.1. Study Area and Datasets

Sub-basins situated over the entire terrestrial Republic of Korea (ranging between latitude 34–39° N and longitude 125–130° E) were mainly focused in this research as the study area. The separation of sub-basins relies on the Korean "Hydrologic Unit Map" (HUM) developed by central ministries such as The Ministry of Land, Infrastructure, and Transport, The Ministry of Agriculture, Food, and Rural Affairs, and The Ministry of Environment. Specifically, the HUM is classified into 5 watersheds covering the 5 largest rivers in Korea (Han River, Nakdong River, Geum River, Seomjin River, and Yeongsan River), 21 large-sized areas (basin level), 117 medium-sized areas (sub-basin level), and 850 small-sized areas (standard basin level). The readers are referred to the Water Resources Management Information System website (WAMIS, http://www.wamis.go.kr/ (accessed on 16 June 2023)) for more details about the HUM. In this study, the evaluation at the sub-basin level (medium size) was considered (Figure 1a). The recent 10-year (2011–2020) daily dataset of the sub-basin averaged runoff, developed from the Korean in situ streamflow network, was selected as the independent reference product to evaluate single and merged reanalysis-based runoff products and can also be obtained from the WAMIS. Moreover, we provided a reference land use land cover map from the Global Consensus Land Cover (GCLC) product [37] for more information on land and vegetation conditions over the study area. To focus on the vegetation effects on runoff behavior, we simplified by grouping different land covers into two major dominant vegetation covers, where shrublands, croplands, grasslands, and barren soils will be classified as low vegetation, and forests will be classified as high vegetation (Figure 1b).

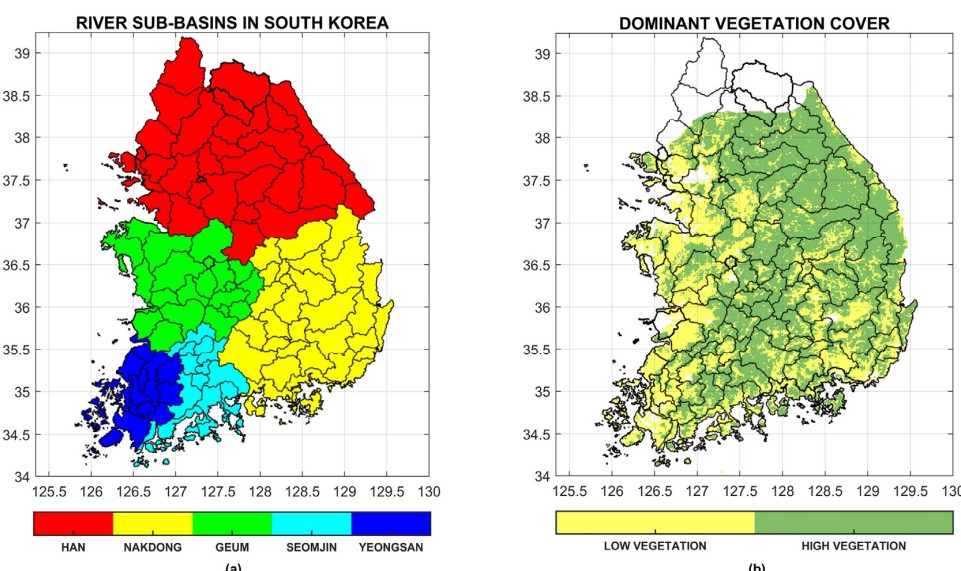

**Figure 1.** Detailed information on the study area of the terrestrial Republic of Korea includes (**a**) Separation of sub-basins for corresponding watersheds over the study area; (**b**) dominant vegetation covers against the sub-basins over the study area (Han River—high vegetation, Nakdong River—low vegetation, Geum River—low vegetation, Seomjin—high vegetation, Yeongsan—low vegetation).

Three commonly used reanalyses, which also experienced multiple careful evaluation processes at the global scale, have been considered to provide standalone runoff products in this study. First, the runoff from the ERA5L [16] is one of the three selected reanalysis datasets. This is a successor of the 5th-generation ECMWF atmospheric reanalysis (ERA5) [15] derived via a re-run of the land components. Original global hourly data at a 0.1° grid during the long-term period (1950-present) of the ERA5L are accessible at the Copernicus Climate Data Store (CDS, 10.24381/cds.e2161bac) [38]. In this study, surface

runoff and baseflow products from the ERA5L covering the subset scene of the Republic of Korea and during the 10-year study period (2011–2020) were extracted. The gridded total runoff product can be considered the sum of surface and baseflow runoff.

Another gridded runoff product used was acquired from NASA GLDAS's latest version, the Catchment Land Surface Model (CLSM) version 2.2 (GLDAS-2.2). This was developed based on data assimilation (DA) of the Princeton meteorological forcing inputs and the novel Gravity Recovery and Climate Experiment (GRACE) satellite data retrieved in the period of 2003-present [39]. In this current research, we employed surface and baseflow runoff from level 4 (L4) daily $0.25° \times 0.25°$ gridded GLDAS CLSM GRACE-DA1 v2.2 data (GLDAS_CLSM025_DA1_D) distributed via the NASA Goddard Earth Sciences Data and Information Service Center (GES DISC, 10.5067/TXBMLX370XX8) [40], with total runoff data as the sum of those both runoff products. Moreover, the subset scene for the Republic of Korea and the study period of 2011–2020 were also extracted from global datasets.

The final reanalysis-based gridded runoff product used in this study was generated from the latest version (version 2) of MERRA global atmospheric reanalysis, the MERRA-2 [41], by the NASA Global Modeling and Assimilation Office (GMAO). An improvement in the new version is enabling DA of modern hyperspectral radiance and microwave satellite observations. Original global hourly time-average MERRA-2 Land Surface Diagnostics data covering the 1980-to-present period at a $0.5° \times 0.625°$ grid (M2T1NXLND) were provided through the NASA GES DISC (10.5067/RKPHT8KC1Y1T) [42]. Similarly, we extracted the surface and baseflow fluxes with respect to the Republic of Korean region subset and a study period of 2011–2020 and summed both of them for the total runoff product. The differences in specifications of three independent reanalysis-based runoff products used in this study are highlighted in Table 1.

**Table 1.** Specification of the original total runoff data (Q) from three selected reanalyses and the methods for resampling original data into the specification of targeted data.

| Reanalysis | Data Type/DOI | Temporal Resolution | Spatial Resolution | Original Unit | Temporal Resampling | Spatial Resampling | Unit Conversion |
|---|---|---|---|---|---|---|---|
| ERA5-Land (ERA5L) | ERA5-Land 10.24381/cds.e2161bac | hourly | $0.10° \times 0.10°$ | m | sum of 24-h data | none | $=(Q/86{,}400) \times$ targeted area |
| GLDAS-2.2 (GLDAS) | GLDAS_CLSM025_DA1_D 10.5067/TXBMLX370XX8 | daily | $0.25° \times 0.25°$ | $kgm^{-2}s^{-1}$ | none | nearest neighbor | $=(Q \times 10^{-3}) \times$ targeted area |
| MERRA-2 (MERRA) | M2T1NXLND 10.5067/RKPHT8KC1Y1T | hourly | $0.50° \times 0.625°$ | $kgm^{-2}s^{-1}$ | sum of 24-h data | nearest neighbor | $=(Q \times 10^{-3}) \times$ targeted area |

*2.2. Methods*

2.2.1. Data Preprocessing

Since the three original gridded reanalysis-based runoff products used in this study are different in measurement space, time, unit, and dynamic ranges, it is necessary to harmonize those datasets matching the specification of the targeted product before applying the TC merging. Although there are differences in runoff behaviors among the five river basins in the Republic of Korea (e.g., the difference in runoff travel time of each river), which can lead to uncertainties when evaluating different river basins, their impacts are not remarkable regarding the entire Republic of Korea and can be negligible in this study. Thus, we decided to select the fixed daily $0.10° \times 0.10°$ runoff data in $m^3 \cdot s^{-1}$ as the targeted specification of the merged product to match the high spatial resolution from the finer scale product (ERA5L) and suitable temporal resolution and unit for the ground-based sub-basin averaged runoff dataset (WAMIS). In particular, a series of preprocessing steps were implemented for the gridded reanalysis-based runoff products including:

1. **Temporal Resampling:** To resample the original hourly runoff dataset to the targeted daily runoff dataset by summing 24-h runoff data in one day. Its application for each single product is mentioned in Table 1.
2. **Spatial Resampling:** To resample the original runoff dataset with a coarser grid to the targeted runoff dataset with a finer grid ($0.10° \times 0.10°$) by using nearest-neighbor

interpolation. Since this simple method can ensure the nature of original datasets without generating any artifacts, it is assumed that the uncertainties associated with this resampling method are small and can be also negligible in this study. Its application for each single product is mentioned in Table 1.

3. **Unit Conversion:** To step-by-step convert the original runoff unit to the pixel runoff depth unit (in $m \cdot s^{-1}$) and then to the targeted runoff unit (in $m^3 \cdot s^{-1}$) by multiplying with the targeted pixel area (supposing that a 0.10° pixel grid ~$10^8$ $m^2$ pixel area). Its application for each single product is mentioned in Table 1.

4. **Seasonal Separation:** This study employed the seasonal merging of runoff products, so separating the seasonal runoff variation periods is required. Despite the four clear seasons in the Republic of Korea, the runoff behavior can vary following the seasonal rainfall variation under the East-Asia monsoon effects rather than these four seasons, where intensified rainfall occurring in the rainy season (mostly summer and fall) [31] can lead to a rapid increase in total runoff. Therefore, we decided to separate the runoff data period in the Republic of Korea into two seasons based on the rainfall seasons. Specifically, the period belonging to the summer and fall seasons (April–September) can be regarded as the wet season, whereas the remaining period can be considered the dry season.

5. **Data Rescaling:** Single datasets need to be rescaled to a similar range before applying the TC merging method. It was suggested that one dataset can be selected as the reference dataset and the two remaining ones will be rescaled to that reference one [43]. In this study, the ERA5L runoff product, which has the highest original spatial resolution, will be assigned as the reference. To ensure that the two remaining datasets can vary within the dynamic range of the reference without generating any invalid values (e.g., negative runoff values), a max-min normalization method will be conducted by matching the normalized values of the remaining datasets to that of the reference. Specifically, we applied this normalization to the GLDAS and MERRA runoff datasets to rescale their data to the ERA5L dynamic range with respect to separated wet and dry seasons as follows:

$$ORG_{norm} = \frac{REF_{max} - REF_{min}}{ORG_{max} - ORG_{min}}(ORG - ORG_{min}) + REF_{min} \tag{1}$$

where *REF* represents the reference runoff dataset (here it was ERA5L), *ORG* represents the original runoff datasets that need to be rescaled (here were GLDAS and MERRA), $ORG_{norm}$ represents normalized/rescaled datasets of *ORG*, and the subscripts max and min stand for maximum and minimum values of the data over a selected period, respectively.

2.2.2. Seasonal Triple Collocation (TC) Merging

The gridded ERA5L, rescaled GLDAS, and rescaled MERRA runoff datasets for the Republic of Korean sub-basins were then used to generate the merged gridded runoff product. The TC merging used as the backbone merging method in this study was developed from the commonly used TC analysis. Specifically, regarding the gridded runoff products, the TC analysis allows the error variances of three independent and collocated measurement systems (here, they are the ERA5L, GLDAS, and MERRA reanalyses) can be characterized without knowing any "truth" measurement, based on the assumption that they are related following the linear additive model [20]:

$$Q_i = \alpha_i + \beta_i q + \varepsilon_i \tag{2}$$

where $Q_i$ (with i = 1, 2, and 3) represents the triplet spatially and temporally collocated reanalysis-based runoff datasets; $\alpha_{\neg I}$ and $\beta_i$ represent systematic additive and multiplicative biases in the runoff dataset i with respect to the true runoff dataset; *q* is the unknown true runoff dataset; and $\varepsilon_i$ is the zero-mean random noises. Hereafter, we use subscripts 1,

2, and 3 to denote the ERA5L, rescaled GLDAS, and rescaled MERRA runoff products, respectively.

In the TC analysis, the error variance maps of triplet reanalysis-based runoff datasets ($\sigma_1{}^2$, $\sigma_2{}^2$, and $\sigma_3{}^2$) can be computed using the covariance notation ($\sigma$) as [21,43]:

$$
\begin{aligned}
\sigma_1^2 &= \sigma_{11} - ((\sigma_{12}\sigma_{13})/\sigma_{23}); \\
\sigma_2^2 &= \sigma_{22} - ((\sigma_{12}\sigma_{23})/\sigma_{13}); \\
\sigma_3^2 &= \sigma_{33} - ((\sigma_{13}\sigma_{23})/\sigma_{12})
\end{aligned}
\tag{3}
$$

where $\sigma_{ij}$ (with i, j = 1, 2, 3 and i $\neq$ j) represents the covariances between two corresponding reanalysis-based runoff datasets i and j. It is worth noticing that negative error variance values can occur in some cases when applying the TC analysis. Therefore, we marked the pixels with negative error variance values as invalid pixels.

Relying on the computed triplet error variances using the TC analysis, the calculation of weight for each single gridded reanalysis runoff product ($w_1$, $w_2$, and $w_3$) in the TC merging method can be conducted as follows:

$$
\begin{aligned}
w_1 &= \frac{\sigma_2^2.\sigma_3^2}{\sigma_1^2.\sigma_2^2+\sigma_1^2.\sigma_3^2+\sigma_2^2.\sigma_3^2}; \\
w_2 &= \frac{\sigma_1^2.\sigma_3^2}{\sigma_1^2.\sigma_2^2+\sigma_1^2.\sigma_3^2+\sigma_2^2.\sigma_3^2}; \\
w_3 &= \frac{\sigma_1^2.\sigma_2^2}{\sigma_1^2.\sigma_2^2+\sigma_1^2.\sigma_3^2+\sigma_2^2.\sigma_3^2}
\end{aligned}
\tag{4}
$$

However, for invalid pixels (negative error variance values), the corresponding weight pixels are also unavailable. To ensure that there are no spatial gaps in the weight maps as well as the merged data, the merging method, in this case, will be changed into the combination of two remaining products. For example, supposing that a pixel in dataset 3 is an invalid pixel, its weight value will be assigned as $w_3$ = 0, and the weights of the corresponding pixels in the two remaining datasets ($w_1$ and $w_2$) can be calculated as [25]:

$$
w_1 = \frac{\sigma_2^2}{\sigma_1^2 + \sigma_2^2}; \; w_2 = \frac{\sigma_1^2}{\sigma_1^2 + \sigma_2^2}
\tag{5}
$$

Finally, the merged gridded runoff product ($Q_{\text{merge}}$) can be computed from the three single reanalysis-based runoff products using the least-square optimal merging as:

$$
Q_{merge} = w_1 Q_1 + w_2 Q_2 + w_3 Q_3
\tag{6}
$$

In this study, the application of the above TC merging procedure for the study period can be regarded as all-time merging, while that for separately wet and dry seasons can be regarded as seasonal merging.

### 2.2.3. Evaluation and Comparison

Evaluation and comparison of different single and merged reanalysis-based gridded runoff products were conducted over the terrestrial Republic of Korean sub-basins against a standalone ground-based runoff product provided by WAMIS. Therefore, the calculation of averaged runoff data for each sub-basin from used reanalyses is required. In particular, the reanalysis data pixels located within the boundary of a sub-basin can be selected as the respective sub-basin pixels. The reanalysis-based sub-basin-averaged runoff data for a given sub-basin can be computed by taking the spatial average of runoff values from all corresponding sub-basin pixels. However, one problem is that some sub-basins are located outside the terrestrial land (e.g., Jeju island), relatively small (covering less than 5 pixels), or have no data (covering invalid pixels or missing value pixels), which are not necessary for this study. A filter to remove the unnecessary sub-basins will be conducted before applying evaluation. We then assess the reanalysis-based sub-basin-averaged runoff data against the

corresponding ground-based sub-basin averaged runoff data during a similar period using the three common statistical evaluation metrics as follows:

**Pearson's Correlation Coefficient (R):**

$$R = \frac{\sum_{i=1}^{N} \left( RA_i - RA_{avg} \right) \left( GR_i - GR_{avg} \right)}{\sqrt{\sum_{i=1}^{N} \left( RA_i - RA_{avg} \right)^2} \sqrt{\sum_{i=1}^{N} \left( GR_i - GR_{avg} \right)^2}} \tag{7}$$

**Unbiased Root-Mean-Square Error (*ubRMSE*):**

$$ubRMSE = \sqrt{(1/N)\sum_{i=1}^{N} \left( \left( RA_i - RA_{avg} \right) - \left( GR_i - GR_{avg} \right) \right)^2} \tag{8}$$

**Mean Bias Error (*MBE*):**

$$MBE = (1/N)\sum_{i=1}^{N} (RA_i - GR_i) \tag{9}$$

where $N$ is the number of samples within the selected period, and $RA_i$ and $GR_i$ are reanalysis- and ground-based sub-basin-averaged runoff data at the ith sample, respectively. Products with higher $R$ values together with lower *ubRMSE* and *MBE* values can be regarded as superior products. Moreover, the positive or negative values of *MBE* can indicate that the reanalysis-based product overestimates or underestimates the ground-based product, respectively. In addition, two comparison schemes were considered and discussed in this study including:

**Comparison of the all-time merged runoff product and the single runoff products:** To investigate whether the output of the TC merging can improve the single parent products at the sub-basin level of the Republic of Korea.

**Comparison of the all-time merged runoff product and the seasonal merged runoff product:** To investigate whether the seasonal TC merging improves traditional all-time TC merging at the sub-basin level of the Republic of Korea.

### 3. Results

*3.1. Error Characterization of Single Reanalysis-Based Runoff Products*

Three standalone reanalysis-based gridded runoff products including the ERA5L, together with the GLDAS and MERRA rescaled to ERA5L product (using Equation (1)), were used as the inputs for applying the TC analysis over the Republic of Korea. The maps in Figure 2 illustrate the spatial pattern of error variances ($\sigma^2$) computed from the TC analysis (Equation (3)) for the entire study period (all-time) and separated wet and dry seasons. Although different spatial $\sigma^2$ patterns were generally observed for distinct reanalysis-based runoff products over sub-basins in the Republic of Korea, a mutual pattern is that the west (west coast of Han River watershed) and south coast (south coast of Nakdong River watershed) regions in the Republic of Korea experienced large $\sigma^2$ values, while the central (Geum River watershed) and southwest (Yeongsan River watershed) regions indicate the lower uncertainties. Among those gridded runoff datasets, the superior product with the lowest $\sigma^2$ values was mainly associated with the ERA5L, followed by the GLDAS and MERRA products. However, in the case of the GLDAS product, several invalid pixels still occurred due to the negative $\sigma^2$ values in this dataset, mainly over the Seomjin River watershed. When it comes to the comparison of wet and dry seasons, significant differences with higher $\sigma^2$ values during the wet season were reported, mainly over the north, east coast, and south areas of the Republic of Korea (covering the Han and Nakdong River watersheds). Regarding the wet season, a complementary $\sigma^2$ pattern has been observed among the runoff products especially over the central Republic of Korea, when the GLDAS and MERRA show high errors but the ERA5L possibly compensates those two products with less uncertainties in this region.

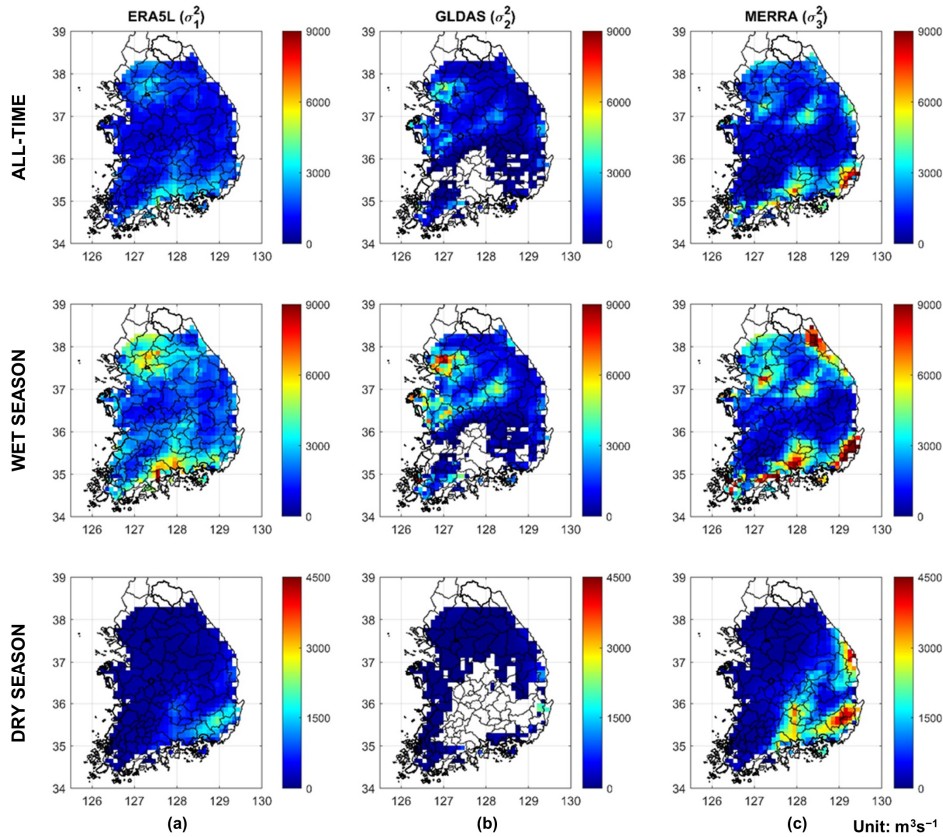

**Figure 2.** Geospatial maps of error variances ($\sigma^2$) from triple collocation analysis during the all-time period, wet, and dry seasons over Republic of Korean sub-basins for three single reanalysis-based runoff products including (**a**) ERA5L ($\sigma_1^2$), (**b**) GLDAS ($\sigma_2^2$), and (**c**) MERRA ($\sigma_3^2$).

### 3.2. Weight Computation for Merging Reanalysis-Based Runoff Products

According to the computed TC-based error variances, the corresponding weights for merging three single reanalysis-based runoff datasets over the Republic of Korea region were then computed using Equation (4) for all-time and separated wet and dry seasons. It is important to note that with respect to invalid pixels in GLDAS $\sigma_2^2$ caused by its negative values as shown in Figure 2, we applied Equation (5) to merge the two remaining runoff products (ERA5L and MERRA). Geospatial maps of those weights (w) against sub-basins in the Republic of Korea are depicted in Figure 3. In general, the spatial variation of w during the all-time period shows a similar pattern to that during the wet season. Particularly, among the runoff products, higher w values in the ERA5L product were assigned to the central and south coasts of the Republic of Korea, where the Geum River and Seomjin River watersheds are situated, respectively. In contrast, higher w values in the GLDAS dataset were mainly observed over the Han River watershed and the east coast of the Nakdong River, while those in the MERRA dataset were over the Yeongsan River. However, when it comes to the dry season, most of the highest w were provided by the GLDAS runoff product except for a part of the central Republic of Korea, the Nakdong River and Seomjin River watersheds, where a large number of invalid pixels of GLDAS caused by negative $\sigma^2$ values occurred. In this case, a vast majority of higher w values were assigned for the ERA5L instead, while less contribution of the MERRA was accounted for in further merged gridded runoff product during this season.

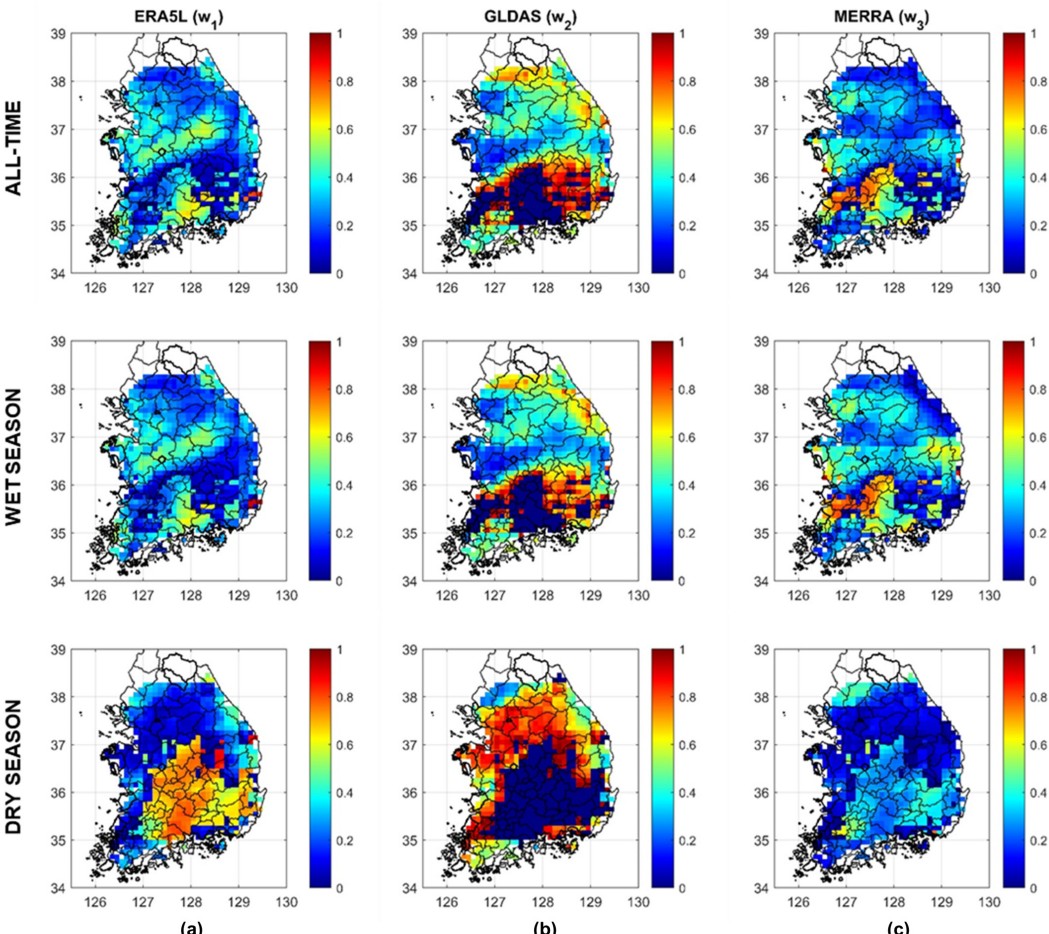

**Figure 3.** Geospatial maps of weights (w) for triple collocation merging during the all-time period, wet, and dry seasons over the Republic of Korean sub-basins for three single reanalysis-based runoff products including (**a**) ERA5L ($w_1$), (**b**) GLDAS ($w_2$), and (**c**) MERRA ($w_3$).

*3.3. Evaluation and Comparison of Merged Gridded Runoff Products*

3.3.1. Comparison of Single and All-time Merged Runoff Products

The three independent reanalysis-based gridded runoff products (ERA5L, GLDAS, and MERRA) and the computed weights (w) in Section 3.2 were then utilized to generate the merged gridded runoff products for the Republic of Korean region following Equation (6). We initially calculated the merged runoff product by using the entire 10-year study period datasets, which is the so-called all-time merged runoff product. Figure 4 provides spatial information on the average and standard deviation values for these three parent products (Figure 4a–c) and the all-time merged product (Figure 4d) against the sub-basins in the Republic of Korea. It can be seen from the figures that the spatial distribution of both the average and standard deviation of the merged runoff product generally shows a similar pattern to its three parent products. In particular, the average map of the merged product is relatively similar to that of the ERA5L, which was considered the reference dataset for scaling GLDAS and MERRA products. However, in the case of standard deviation, high values derived from the merged product map seem to be inherited from the GLDAS and MERRA products rather than the ERA5L. Moreover, regarding different river watersheds, the Han River, Seomjin River, and southeast coast of the Nakdong River indicate higher runoff average and standard deviation values compared to those in the Geum River and Yeongsan River watersheds.

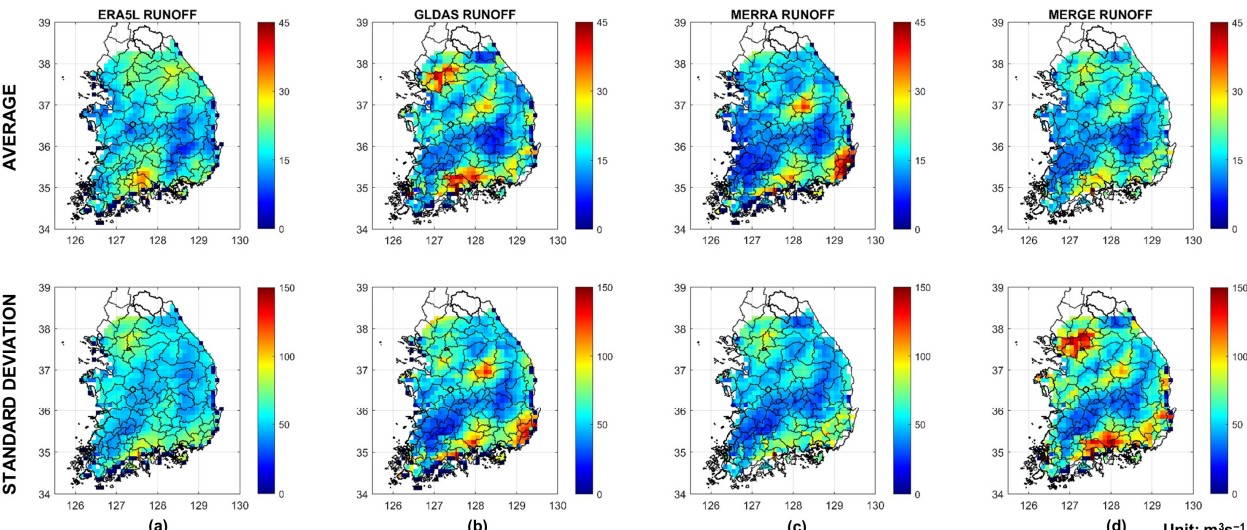

**Figure 4.** Geospatial maps of average and standard deviation for three single reanalysis-based runoff and TC-based merged runoff products during the all-time period over the Republic of Korean sub-basins including (**a**) the ERA5L runoff, (**b**) the GLDAS runoff, (**c**) the MERRA runoff, and (**d**) the MERGE runoff (all-time TC-based merged runoff product).

To investigate the robustness of the TC-based merging method in improving single gridded runoff estimates from reanalysis products, an independent runoff dataset was employed as the benchmark product to assess the quality of the merged runoff product in comparison to its three parent products. Specifically, we compared these products against the ground-based sub-basin averaged runoff product from the Republic of Korea WAMIS site based on the three evaluation metrics including R, ubRMSE, and MBE. Boxplots in Figure 5 represent the evaluation results for all sub-basins in each river watershed, whereas their average values are summarized in Table 2. As can be drawn from Figure 5 and Table 2, among the single parent products, superior performance was observed with the ERA5L and MERRA products, while the GLDAS showed the worst performance. In general, the merged product outperformed its parent products over all five Republic of Korea big river watersheds, with a medium improvement obtained in terms of correlation R and a slight improvement obtained in terms of ubRMSE and MBE. Nevertheless, as regards the MBE results, most of the products revealed negative values, which implies that the reanalysis-based runoff products often underestimate the ground-based product. When comparing the runoff products over watersheds, the Han River, Geum River, and Seomjin River indicate higher R values, with reasonable average values of around 0.60. However, the highest ubRMSE and MBE values, which are the indicators for errors and biases, were also obtained in the Han River watershed, followed by the Nakdong River watershed, with average (ubRMSE; MBE) values of (95.75 $m^3 \cdot s^{-1}$; $-15.23$ $m^3 \cdot s^{-1}$) and (64.36 $m^3 \cdot s^{-1}$; $-3.33$ $m^3 \cdot s^{-1}$), respectively. The Geum River and Seomjin River watersheds, which are located in the central region and south coast of Republic of Korea, are the two regions where the merged runoff product showed superiority in all three evaluation metrics when the highest averaged R values (0.61 and 0.60) and medium to low averaged ubRMSE (45.46 $m^3 \cdot s^{-1}$ and 55.45 $m^3 \cdot s^{-1}$) and averaged MBE ($-4.18$ $m^3 \cdot s^{-1}$ and 2.20 $m^3 \cdot s^{-1}$) were gained.

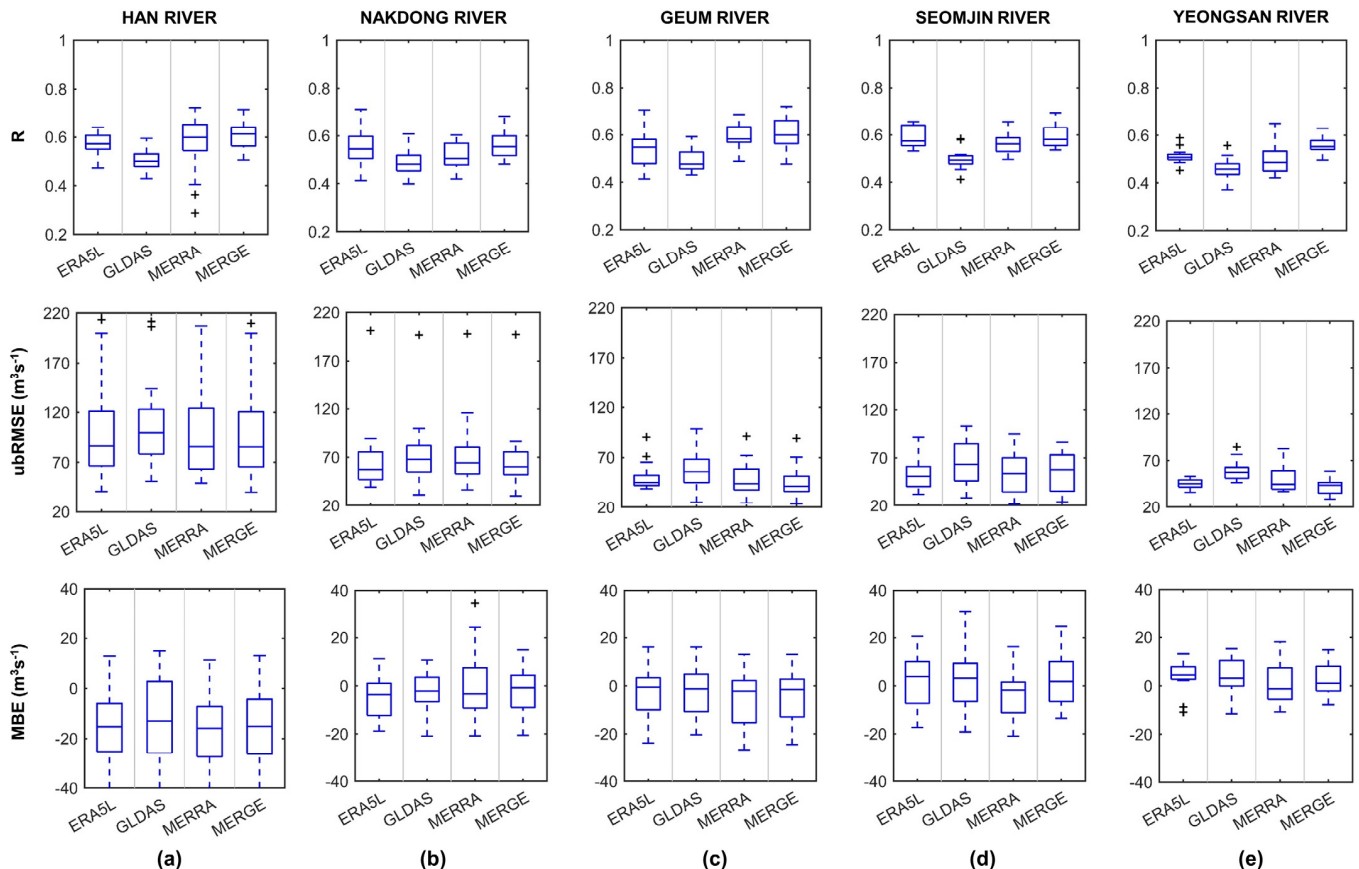

**Figure 5.** Boxplots representing evaluation results (in R, ubRMSE, and MBE) for three single and one all-time merged reanalysis-based runoff products (ERA5L, GLDAS, MERRA, and MERGE) regarding all sub-basins within the 5 river watersheds in the Republic of Korea including (**a**) Han River, (**b**) Nakdong River, (**c**) Geum River, (**d**) Seomjin River, and (**e**) Yeongsan River. The symbol "+" represents an outlier point.

**Table 2.** Watershed averaging evaluation results (in R, ubRMSE, and MBE) for three single and one all-time merged reanalysis-based runoff products (ERA5L, GLDAS, MERRA, and MERGE) regarding all sub-basins within the 5 river watersheds in the Republic of Korea.

| Evaluation | Metric | Reanalysis | Han River | Nakdong River | Geum River | Seomjin River | Yeongsan River |
|---|---|---|---|---|---|---|---|
| Watershed Averaging | R | ERA5L | 0.57 | 0.55 | 0.54 | 0.59 | 0.51 |
| | | GLDAS | 0.50 | 0.49 | 0.49 | 0.50 | 0.46 |
| | | MERRA | 0.58 | 0.52 | 0.60 | 0.59 | 0.56 |
| | | MERGE | 0.61 | 0.56 | 0.61 | 0.60 | 0.57 |
| | ubRMSE $(m^3 \cdot s^{-1})$ | ERA5L | 98.39 | 64.52 | 50.44 | 52.95 | 45.34 |
| | | GLDAS | 107.00 | 71.71 | 56.42 | 65.28 | 59.14 |
| | | MERRA | 97.49 | 72.15 | 47.99 | 53.53 | 50.83 |
| | | MERGE | 95.75 | 64.36 | 45.46 | 55.45 | 41.92 |
| | MBE $(m^3 \cdot s^{-1})$ | ERA5L | −14.86 | −5.98 | −2.44 | 2.42 | 3.58 |
| | | GLDAS | −13.54 | −4.15 | −2.83 | 2.88 | 4.15 |
| | | MERRA | −17.02 | −2.19 | −5.58 | −3.83 | 1.17 |
| | | MERGE | −15.23 | −3.33 | −4.18 | 2.20 | 2.26 |

The time series of three single reanalysis-based runoff products and their all-time merged product at a representative sub-basin in each watershed were extracted and plotted against the corresponding ground-based sub-basin-averaged runoff dataset (WAMIS) in

Figure 6, together with their respective scatter plots, whereas their evaluation results are provided in Table 3. According to Figure 6, the variation in the merged runoff time series highly responded to the variation of the three parent products and the ground-based runoff product at the sub-basin level, especially for the wet season. However, the scatter plots revealed that all the gridded runoff products underestimated the ground-based runoff at the representative sub-basins located in the Han River, Nakdong River, and Seomjin River watersheds, while they overestimated the ground-based product over the representative sub-basins in the Geum River and Yeongsan River watersheds. Results from Table 3 indicate the consistency with those from Table 2 when the merged runoff product outperformed all three original reanalysis-based runoff products, especially in terms of R. Across the watersheds, the highest R values ranging from 0.70 to 0.72 were found in the Han River, Geum River, and Seomjin River watersheds. However, in terms of ubRMSE and MBE, the merged runoff product did not improve the parent products in all cases. The ubRMSE results obtained with the merged product showed a relatively equal over most of the representative sub-basins, except one from the Yeongsan River watershed with the smallest value of 30.88 $m^3 \cdot s^{-1}$. When it comes to the MBE, the highest negative values were observed with the Nakdong River ($-52.54$ $m^3 \cdot s^{-1}$) and Han River ($-22.25$ $m^3 \cdot s^{-1}$) watersheds.

**Table 3.** Representative sub-basin evaluation results (in R, ubRMSE, and MBE) for three single and one all-time merged reanalysis-based runoff products (ERA5L, GLDAS, MERRA, and MERGE) regarding a representative sub-basin in each river watershed in the Republic of Korea.

| Evaluation | Metric | Reanalysis | Han River (ID 1018) | Nakdong River (ID 2018) | Geum River (ID 3002) | Seomjin River (ID 4009) | Yeongsan River (ID 5006) |
|---|---|---|---|---|---|---|---|
| A Representative Sub−Basin in Watershed | R | ERA5L | 0.58 | 0.60 | 0.56 | 0.63 | 0.53 |
| | | GLDAS | 0.60 | 0.56 | 0.59 | 0.58 | 0.56 |
| | | MERRA | 0.72 | 0.58 | 0.68 | 0.60 | 0.53 |
| | | MERGE | 0.72 | 0.67 | 0.72 | 0.70 | 0.63 |
| | ubRMSE ($m^3 \cdot s^{-1}$) | ERA5L | 56.78 | 55.19 | 64.11 | 51.19 | 39.29 |
| | | GLDAS | 82.33 | 69.04 | 64.88 | 70.65 | 56.51 |
| | | MERRA | 56.58 | 63.41 | 61.86 | 45.00 | 36.76 |
| | | MERGE | 55.39 | 58.68 | 60.11 | 61.97 | 30.88 |
| | MBE ($m^3 \cdot s^{-1}$) | ERA5L | −24.48 | −51.61 | 16.15 | −9.65 | 5.12 |
| | | GLDAS | −9.58 | −47.48 | 7.64 | −6.52 | 8.66 |
| | | MERRA | −27.67 | −52.46 | 7.35 | −16.40 | 2.60 |
| | | MERGE | −22.25 | −52.54 | 9.54 | −13.43 | 3.14 |

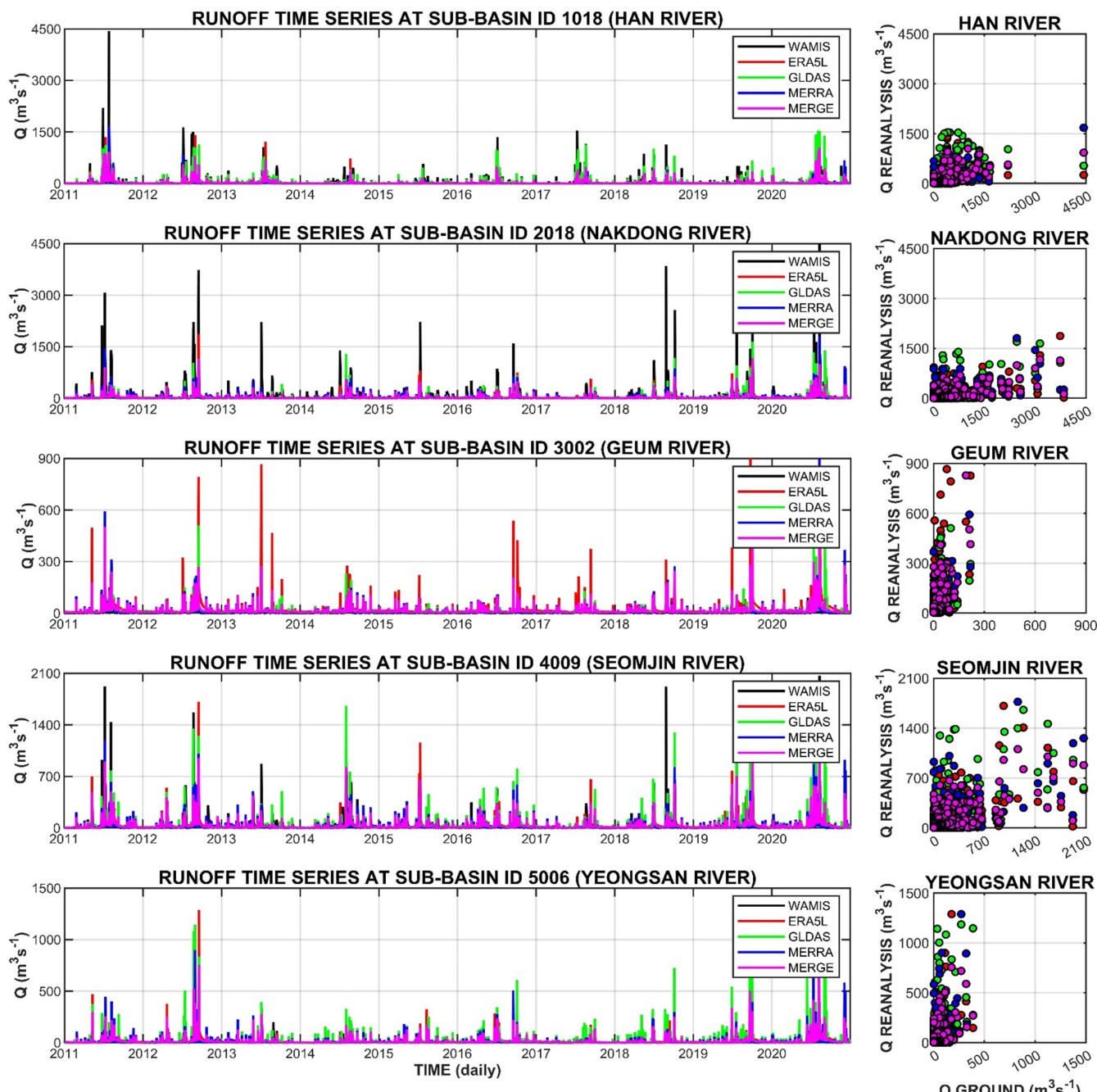

**Figure 6.** Time series and scatter plots of reanalysis-based sub-basin average runoff products (ERA5L, GLDAS, MERRA, and MERGE) against a ground-based sub-basin average runoff product (from WAMIS) at a representative sub-basin in each river watershed.

3.3.2. Comparison of All-time and Seasonal Merged Runoff Products

Figure 7 depicts the geospatial average and standard deviation values of the two merged runoff products generated from the all-time and seasonal TC merging against the Republic of Korean sub-basins, together with their absolute difference values. In general, both merged runoff products indicated a similar spatial variation pattern, with no significant differences observed between these two products. Nevertheless, when we strongly focus on the difference maps (Figure 7c), it can be seen that there are clear differences between them. In particular, larger differences in average values were reported over the northern part, the east and south coasts of the Republic of Korea, where the Han

River, Nakdong River, and a part of the Geum and Seomjin River watersheds are located. Additionally, as regards the standard deviation values, higher differences can be found in the east and south coasts of the Republic of Korea.

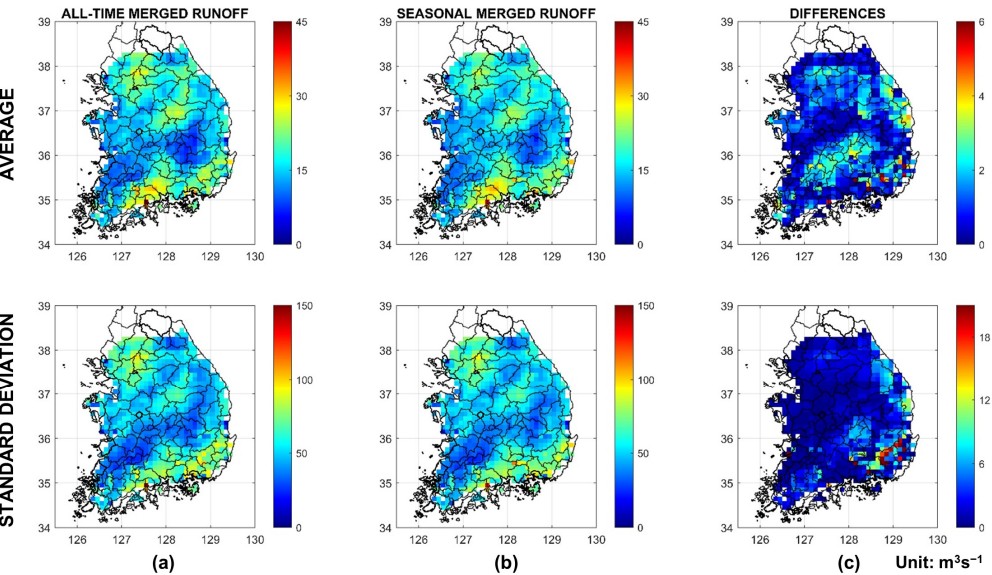

**Figure 7.** Geospatial maps of averages, standard deviations, and differences between two merged runoff products during the all-time period over Republic of Korea sub-basins including (**a**) the all-time TC-based merged runoff product, (**b**) the seasonal TC-based merged runoff product, and (**c**) differences between two merged products (seasonal merged runoff minus all-time merged runoff).

The comparison of outputs from the all-time merging and seasonal merging was also conducted against the ground-based runoff product from WAMIS at the sub-basin level based on the three evaluation metrics. The boxplots in Figure 8 illustrate the comparison results in terms of R, ubRMSE, and MBE for the two merged runoff products over five big watersheds in the Republic of Korea, and their average values are presented in Table 4. As can be drawn from Figure 8 and Table 4, although no significant change in the evaluation results between the two merged runoff products, a slight improvement was obtained with the seasonal merging method considering three evaluation metrics. Particularly, the seasonal merging method markedly improved the all-time merging method in terms of bias; however, underestimation of the ground-based runoff product still occurred. Across different river watersheds, on the one hand, the highest correlation (R) value of 0.63 from the seasonally merged product was reported in the Han River watershed. On the other hand, this region also experienced the largest error (ubRMSE, 94.75 $m^3 \cdot s^{-1}$) and bias (MBE, $-15.83$ $m^3 \cdot s^{-1}$) values, even in the seasonal merging's output.

**Table 4.** Watershed averaging evaluation results (in R, ubRMSE, and MBE) for two merged reanalysis-based runoff products (all-time merged and seasonal merged products) regarding all sub-basins within the 5 river watersheds in the Republic of Korea.

| Evaluation | Metric | Merged Reanalysis-Based Runoff | Han River | Nakdong River | Geum River | Seomjin River | Yeongsan River |
|---|---|---|---|---|---|---|---|
| Watershed Averaging | R | All-time Merging | 0.61 | 0.56 | 0.61 | 0.60 | 0.57 |
| | | Seasonal Merging | 0.63 | 0.57 | 0.61 | 0.61 | 0.58 |
| | ubRMSE ($m^3 \cdot s^{-1}$) | All-time Merging | 95.75 | 64.36 | 45.46 | 55.45 | 41.92 |
| | | Seasonal Merging | 94.75 | 65.77 | 45.59 | 53.60 | 41.02 |
| | MBE ($m^3 \cdot s^{-1}$) | All-time Merging | $-15.23$ | $-3.33$ | $-4.18$ | 2.20 | 2.26 |
| | | Seasonal Merging | $-15.83$ | $-2.78$ | $-3.70$ | 2.40 | 1.41 |

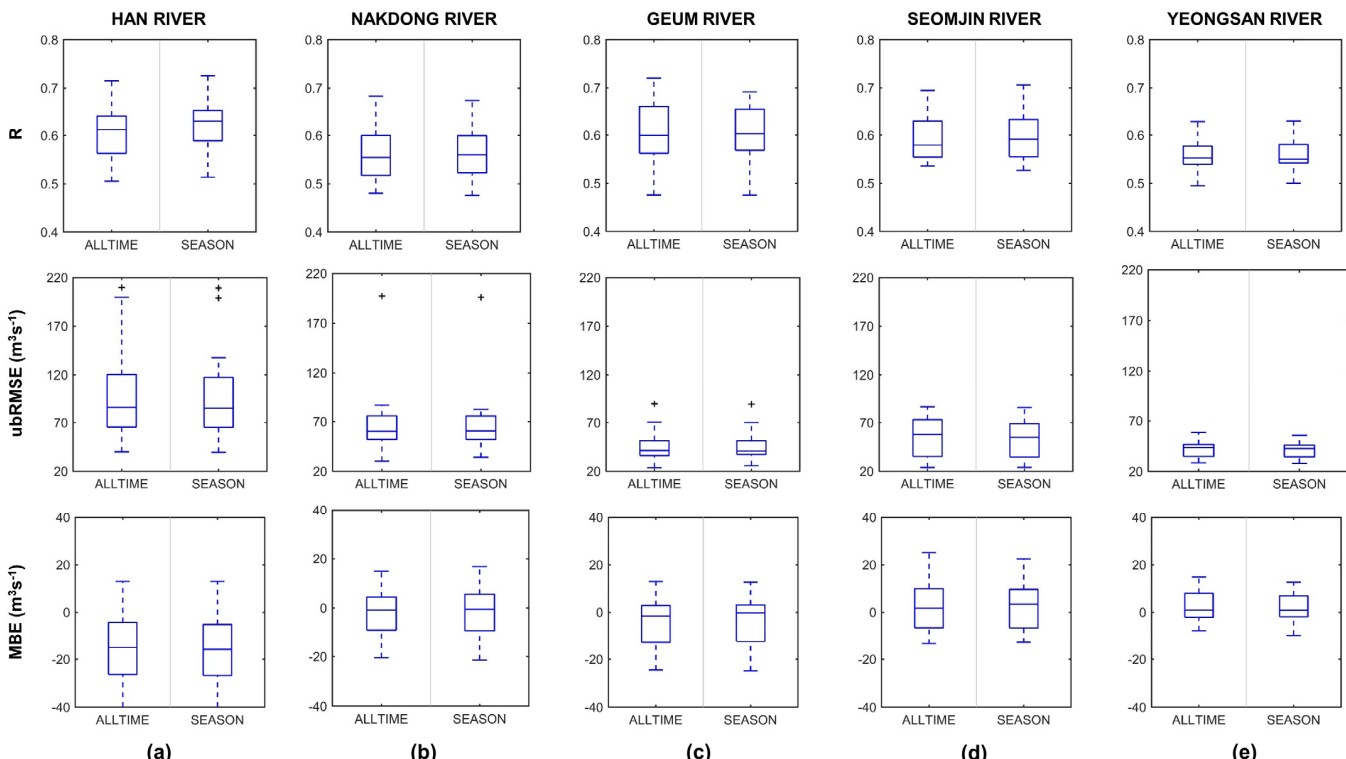

**Figure 8.** Boxplots representing evaluation results (in R, ubRMSE, and MBE) for two merged reanalysis-based runoff products (all-time merged and seasonal merged products) regarding all sub-basins within the 5 river watersheds in the Republic of Korea including (**a**) Han River, (**b**) Nakdong River, (**c**) Geum River, (**d**) Seomjin River, and (**e**) Yeongsan River. The symbol "+" represents an outlier point.

## 4. Discussion

Higher uncertainties in runoff datasets, expressed by the higher error variances observed in wet seasons are in line with the high spatiotemporal rainfall variability during this season, which can make these datasets more likely non-stationary when the all-time period is considered. Moreover, the results from the comparison of three single reanalysis-based gridded runoff products against the ground-based sub-basin average runoff product from the WAMIS over five big watersheds in the Republic of Korea revealed the superior performance of the ERA5L compared to the two remaining products. This could be because the ERA5L was used as the reference for scaling the GLDAS and MERRA runoff datasets in this study. Moreover, the ERA5L reanalysis, with its highest spatial resolution (0.10° grid) among the three reanalysis datasets, can capture the spatial variation of meteorological forcing variables as well as the runoff in the Republic of Korea. This result is in line with the finding from several previous studies conducted over East Asia [29,44] when the outputs from ERA5L reanalysis showed the best agreement with the ground-based measurements.

As regards the comparison of the performances between the all-time TC merging and the single products, the fact that the merged runoff dataset outperformed its single parent datasets highlighted the necessity for combining multiple gridded runoff products to overcome the limitations associated with single products over the Republic of Korean sub-basins and the suitability of applying the TC merging method for improving runoff estimates in this region. However, negative bias (MBE) values were observed in the merged runoff product, which was inherited from the single parent products, implying the underestimation of these reanalysis outputs over the ground-based runoff measurement. This underestimation was also found in previous studies [16,41,45–48], suggesting further application of the bias correction methods for the merged product before putting it into

hydrologic models. Considering the different watersheds, superior performances in terms of three evaluation metrics can be observed in the central and south coasts of the Republic of Korea, where the Geum River and Seomjin River watersheds are situated. This is due to the fact that higher weights were assigned to the most superior product in these regions, the ERA5L runoff product. Especially for the central Republic of Korea, the better performance of the ERA5L runoff also links to the superior performance of the ERA5L rainfall product in this region, which is consistent with the previous finding [31,49]. Moreover, according to the vegetation cover map in Figure 1b, it can be seen that superior performances of the merged product over the Geum River, Seomjin River, and Yeongsan River (in terms of lowest ubRMSE and MBE values) watersheds overlap a vast majority of low vegetation regions. Various studies demonstrated that the increase in vegetation density or densely forested areas can significantly reduce surface runoff rather than low vegetation covers such as grasslands or croplands [50–52], which may result in a longer response time between rainfall and runoff and decrease the runoff estimation performance at short-time-interval (e.g., on a daily basis). On the other hand, higher errors (in ubRMSE) were found in the two biggest watersheds in the Republic of Korea, the Han River and Nakdong River watersheds, implying high heterogeneity at the sub-basin level over the large areas. This result suggests that further evaluations at a smaller level (e.g., station level) need to be considered in these regions.

Superior performance obtained with the seasonal merging method compared to the all-time merging method demonstrated the robustness of this modification over the traditional TC approach for runoff, which has a clear seasonal pattern caused by seasonal rainfall effects. In particular, for variables with seasonal behaviors, several assumptions in the TC analysis, especially the stationarity of the time series data, are often not strictly followed during the application of TC analysis [43]. By separately considering different seasonal periods for wet and dry seasons, we can remove the seasonality during the entire period, which can make the aforementioned non-stationarity problem induced by seasonal time series become more stationarity, improving the effectiveness of the TC merging method. Although only a slight improvement was reported, the seasonal TC merging used in this study still shows a potential application when shorter seasonal periods and detailed seasonal behaviors are considered.

## 5. Conclusions

This study aims to improve the inherent uncertainties in single reanalysis-based runoff products over the heterogeneous Republic of Korean region by merging multiple global reanalyses to generate a high-quality and long-term gridded runoff product at a high resolution. The commonly used triple collocation (TC) analysis, which allows the combination of multiple standalone and collocated datasets without any requirement of truth reference datasets, was employed as the backbone of the merging method. In particular, we synthesized three global reanalysis-based runoff products including the ERA5L, GLDAS, and MERRA during a 10-year study period to produce a daily merged dataset at 0.1° grid. Moreover, a modification of separately applying the TC merging for wet and dry seasons (seasonal merging) was also conducted in this study, expecting to enhance the seasonal runoff estimates induced by seasonal rainfall variation in the Republic of Korea. The performances of those reanalysis-based gridded runoff products were compared at the sub-basin level against an independent ground-based sub-basin averaged runoff product distributed from the local WAMIS network.

The results of the TC-based error variances ($\sigma^2$) and weights (w) revealed that lower $\sigma^2$ values together with higher w values in the ERA5L product were observed over the central Republic of Korea (covering the Geum River watershed) and the south coast of the Republic of Korea (covering the Seomjin River watershed), while those of GLDAS and MERRA products were found over the three remaining watersheds.

The results when comparing the single runoff products to the all-time merged runoff product indicated that the TC-based merged product generally outperformed its parent

products, highlighting the robustness of applying the TC merging method to improve runoff estimates over Republic of Korean sub-basins. Among the watersheds, superior performances can be observed in the Geum River and Seomjin River watersheds because higher weights were assigned to the most superior product (the ERA5L) in these regions. In contrast, higher errors and biases were found in the two biggest watersheds in the Republic of Korea, the Han River and Nakdong River watersheds, implying high heterogeneity at the sub-basin level over the large areas, and this may require further detailed evaluation at the station level and prior bias correction application.

The comparison results between the all-time and seasonal merging methods demonstrated the enhancement of the seasonal merging product, especially in terms of bias. This could be because the non-stationary runoff problem caused by seasonal rainfall effects in the Republic of Korea, which is one of the major drawbacks in the current TC assumption, can be addressed by removing the seasonality based on separately considering different seasonal periods. Although only a slight improvement was reported, the seasonal TC merging used in this study still shows a potential application for boosting the TC merging method when shorter seasonal periods and detailed seasonal behaviors are considered.

This is a preliminary study toward the construction and application of a long-term and high-quality gridded runoff product for the Republic of Korea region, so several limitations still remain. Future studies will focus on addressing the current drawbacks such as evaluation, comparison, or bias correction at high-resolution grid-scale or station level; and effectively applying this high-quality runoff dataset as a benchmark product for developing artificial intelligence-based hydrologic models or improving flood/drought early warning systems.

**Author Contributions:** Conceptualization, W.-Y.S.; methodology, W.-Y.S. and H.H.N.; software, W.-Y.S. and H.H.N.; validation, W.-Y.S. and H.H.N.; formal analysis, W.-Y.S. and H.H.N.; investigation, W.-Y.S. and H.H.N.; resources, W.-Y.S.; data curation, W.-Y.S.; writing—original draft preparation, W.-Y.S.; writing—review and editing, H.H.N.; visualization, W.-Y.S. and H.H.N.; supervision, K.-S.J.; project administration, W.-Y.S. and K.-S.J.; funding acquisition, K.-S.J. All authors have read and agreed to the published version of the manuscript.

**Funding:** This research was funded by the Korea government (MSIT), supported by the National Research Foundation of Korea (NRF), grant number NRF-2021R1A6A3A01087386; and the Korea Environmental Industry and Technology Institute (KEITI), grant number 2022003460001. The APC was funded by NRF-2021R1A6A3A01087386 and KEITI-2022003460001.

**Data Availability Statement:** The data presented in this study are available upon request from the corresponding author.

**Acknowledgments:** The authors acknowledge the ECMWF and CDS for providing free access to the ERA5-Land dataset; and the NASA GES DISC for providing free access to GLDAS-2.2 and MERRA-2 datasets used in this study. We would like to thank the Department of Ecology and Evolutionary Biology at Yale University, USA, for the GCLC dataset. Finally, we are grateful to the Water Resources Management Information System (WAMIS), Republic of Korea, for providing the sub-basin level runoff dataset.

**Conflicts of Interest:** The authors declare no conflict of interest. The funders had no role in the design of the study; in the collection, analyses, or interpretation of data; in the writing of the manuscript; or in the decision to publish the results.

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
