# Peer review of "Generation of High-Resolution Gridded Runoff Product for the Republic of Korea Sub-Basins from Seasonal Merging of Global Reanalysis Datasets"

_water, doi:10.3390/w15213741_

Round 1

Reviewer 1 Report

Comments and Suggestions for Authors

1. The author selected three grid-based runoff products with significant differences in spatial resolution (0.1°x0.1°, 0.25°x0.25°, 0.50°x0.625°),and used nearest neighbor interpolation method to resample the low-resolution data to a higher resolution. Does this approach have an impact on the final fusion results? Additionally, there are multiple grid-based runoff products available, so why choose these three? It is recommended to supplement the introduction in Section 2.1 by highlighting their advantages and explaining the reasons for selecting them in this study.

2. The quality comparison of the runoff products is conducted at the subbasin scale, without examining the effect at the grid scale. Therefore, for the merged product with a resolution of 0.10° or other resolutions, there is no fundamental difference. One of the goals of the paper is to generate "high-quality and long-term high-resolution gridded runoff products," which can hardly be considered reliable high-resolution products without evaluating the performance at the grid scale.

3. Since seasonal merging is already being pursued, I would be pleased to see the authors attempt to directly achieve resolution at the four seasonal or even monthl.

4. The visualization in Figure 6 is not effective, making it difficult to distinguish among the five runoff products. It is recommended that the authors consider changing the presentation method for this figure.

5. Since deviations have been identified in the merged product, why not utilize WAMIS for subbasin-scale correction and generate a truly "high-quality gridded runoff product"?

6. The conclusion section contains a lot of sentence repetition, restating the earlier content. It is suggested that the authors make the conclusion section more concise and logically structured.

Reviewer 2 Report

Comments and Suggestions for Authors

This manuscript is well-organized. The methodology is detailed. The obtained results are carefully discussed and properly illustrated. After the necessary revisions, I feel this manuscript can be approved.

How does the developed runoff product account for processes such as evaporation from the catchment surface, soil moisture dynamics, and snow cover in the mountainous region of the country?

The Nash-Sutcliffe NSE and Kling Gupta KGE criterion should be added to the above provided measures for matching hybrid and observed discharges.

There is no information available for the five river basins where the produced product is being verified. It appears that the runoff travel time for some rivers may be shorter than a day. Especially when it comes to small rivers. How did the authors resolve this issue?

Have you evaluated the reproduction of maximum river runoff during the monsoon season?

Reviewer 3 Report

Comments and Suggestions for Authors

The manuscript is focused on the three single reanalysis-based gridded runoff products against the ground-based sub-basin average runoff product from the WAMIS (nationala database) for five big watersheds in South Korea revealed the superior performance of the ERA5L 456 compared to the two remaining products. This could be because the ERA5L was used as the reference for scaling the GLDAS and MERRA runoff datasets in this study.

Methodology is based on using the geospatial statistical tools and represent the standard tools using in GIS analyses. The results are clearly presented in statistical as well as map graphics point of view. The conclusions are supported by the presented results.

I have no further comments or recommendations.

Author Response

Thank you very much for your comments. We appreciate your time in reviewing this manuscript.

Round 2

Reviewer 1 Report

Comments and Suggestions for Authors

The authors have addressed all the questions I raised, and I have no further concerns.

Author Response

Thank you so much for your kind and valuable comments.